# Culturally Informed Technology: Assessing Its Importance in the Transition to Smart Sustainable Cities

**Ibrahim Mutambik** 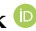

Department of Information Science, College of Humanities and Social Sciences, King Saud University, Riyadh P.O. Box 11451, Saudi Arabia; imutambik@ksu.edu.sa

**Abstract:** Since the idea of the smart city was first introduced, over two decades ago, there has been an increasing focus on sustainability as a core strategic priority. However, as the relevance, importance and even definition of sustainability is a function of cultural context, planners must take account of local and regional cultural factors in the selection and adaption of digital infrastructures, as well as in the management and encouragement of public acceptance. Achieving this is not a sequential process, but a concurrent one, as these factors are interdependent. This raises the question of what factors affect and mediate the technology, choice, and public acceptance of smart sustainable cities. This paper attempts to address this question by proposing a new model which advances our current, and considerable, understanding of Technology Acceptance Modelling—using an analysis based on Structural Equation Modelling. This new model, called the Culturally Informed Technology Acceptance Model, was validated using data from a survey of residents of a variety of Saudi Arabian cities. The proposed model is designed around important factors that can be influenced by cultural context, such as digital literacy, process improvements, cost savings and privacy, and is a useful tool for understanding the role of culture in the public acceptance of smart sustainable technology. This design focus is for a number of reasons, such as helping development bodies ensure that the technologies used align with the socio-cultural context. It will also help in the management of at-scale technology roll out in a way that is resource-efficient. Although the Culturally Informed Technology Acceptance Model has been developed and validated using data from Saudi Arabia, the authors believe that it could be adapted to meet the needs of countries/cities that are looking to implement smart city strategies matched to their own distinct socio-cultural identity.

**Keywords:** citizens' participation; smart cities; sustainability; smart sustainable cities; social capital; technology acceptance model; urban

## 1. Introduction

The concept of the smart city offers many benefits to both governments and their citizens [1–3], and a strategy of smart city transition has been adopted by many countries over the past couple of decades [4–6]. In fact, the smart city market is expected to reach a value of over US$100 bn in 2024, and growth is expected to continue at a CAGR (compound annual growth rate) of 12.15% from 2024, to reach a value of almost $170 bn in 2028 [7] and over $300 bn by 2032 [8]. In more recent years, however, it has been increasingly recognised that the general concept of the smart city has not necessarily reflected the global shift of governmental focus to environmental, social, and economic sustainability [9,10]. This has led to the emergence of the concept of the smart sustainable city, which prioritises sustainability as a core objective, integrating environmental, social, and economic considerations into all aspects of urban planning and development [11].

The successful planning and implementation of smart sustainable cities, however, depends critically on an issue which has, until now, received little attention in the literature. This issue is that the notion of 'sustainability' is culturally dependent, and that, as a result, the cultural identity of a city will influence how residents and organisations approach

and adopt new technologies. A city's cultural identity should therefore inform the overall implementation of smart sustainable city initiatives [12–14]. Despite this, however, the smart sustainable city is typically promoted as a generalised concept that is universally applicable [15,16], and planners tend to treat the transition to a smart sustainable city as the mere application of existing technology in new ways [16]. This study challenges this perspective by examining local and regional factors which affect the engagement and acceptance of smart sustainable technology.

In fact, the impact of culture on the acceptance of smart technology can be considered to be a result of a number of causes. In some cultures, for example, embracing new technology is seen as a sign of progress, while, in others, it might be met with scepticism or resistance [17]. Another issue is trust and familiarity. Societies, that have embraced mobile payment systems, for instance, may be more open to adopting other smart technologies [18,19]. In yet other social contexts, social networks and peer influence play a role: if friends, family, or colleagues adopt a particular technology, others are likely to follow suit [20,21]. Cultural norms around social conformity can also impact technology adoption.

These issues can be illustrated by a number of specific instances of how culture affects smart technology uptake. In Japan, for example, the use of smartphone apps for healthcare management and telemedicine has been significantly shaped by cultural attitudes toward health and technology. The Japanese culture places a strong focus on preventive healthcare and personal wellness, making smartphone apps that track fitness and provide health-related information appealing to users [22]. Another example is in Singapore, a country which has implemented smart transportation systems, including electronic road pricing (ERP) and an extensive network of sensors and cameras for traffic management. The acceptance of these technologies has been influenced by cultural factors such as efficiency, orderliness, and government-led initiatives [23–25]. It is worth noting that some technologies, such as blockchain, are accepted by default, as they are 'invisible'—i.e., they are deployed in a way that is not evident to the user [26,27].

Since the emergence of the digital computer in the 1960s, digital technology has become ever-more integral to global society, such that, by January 2024, 66.2% of the world's population (5.35 billion people) had access to the Internet [28]. However, this access and usage is not evenly distributed [29,30], and some regions of the world are therefore more advanced than others in their implementation of smart city technology. This creates a key challenge for governments: on the one hand, the less advanced regions are those which could potentially benefit most from smart city implementation [31,32], while, on the other hand, it is particularly important, in these areas, to understand the human factors which affect acceptance.

This point underscores the significance of Technology Acceptance Models (TAMs), which play a vital role in successfully forming and deploying smart and sustainable city technologies. Originally introduced by Davis in 1987 [33], and later refined by Davis and Bagozzi [34], the TAM is a theoretical framework developed to understand and predict the acceptance and usage of emerging information technologies. Widely used to study the adoption of technology in different contexts, ranging from e-commerce to healthcare, TAMs can provide valuable insights into the factors that influence user acceptance, and can play an important role in helping to design and develop technologies that are more likely to be accepted and used by their target users. However, such models can be especially useful in contexts such as Saudi Arabia, where smart cities play a major role in the realisation of the Saudi Vision 2030, by contributing to the goals of economic diversification, technological innovation, sustainability and improved quality of life [35–38]. Although the majority of the Saudi population is Arab, there is cultural diversity in terms of regional identities, traditions and practices [35], which makes the TAM particularly relevant, as our understanding of the effects and roles of culture in technology adoption remains limited [39].

The principal aim of this study is to help smart sustainable city planners and strategists, as well as industries and businesses, understand the importance of 'Culturally Informed' technologies which are appropriate for the local context. These technologies will form the basis of a range of public services [40,41], which citizens can easily engage with and benefit from [42,43]. Such services, which lie at the heart of life in a smart city, whether focused on sustainability or not, take a wide variety of forms, ranging from intelligent traffic management and smart parking apps, to remote healthcare consultations and delivery [40,44]. However, as cultural nuances can significantly impact preferences and expectations, ensuring that technologies are selected and deployed in a way that resonates with the target audience, especially in terms of its perception of the importance of sustainability, will help to maximise the acceptance and adoption of smart sustainable public services [45,46].

The rest of this document is structured in the following manner: Section 2 offers a concise review of the literature relevant to this research, while Section 3 presents the hypotheses that will be tested. Section 4 outlines the research methods and the data collection process, and Section 5 presents the results of the data analysis and the consequent validity of the study's hypotheses. Section 6 discusses the findings and the main implications for further research. Section 7 concludes the paper, by highlighting the study's limitations and suggesting directions for future research.

## 2. Literature Review

It is interesting to note that, over the past twenty years or so, there has been an almost total inversion of the perception of the role of regionality in the development of smart cities. At one point, in the early 2000s, the smart city was considered a generic, and universally applicable, concept that was defined only by technology, rather than by geographical or cultural context [29,30]. In more recent years, however, as the global understanding of smart technologies has matured, and the idea of the smart sustainable city has begun to take shape, the criticality of local factors in the development of smart cities has begun to be recognised [47,48]. The social, economic and sustainability benefits of powerful and emerging communications technologies have been shown to vary according to geography and regional culture [49,50], and it has also been shown that smart services have higher rates of adoption when aligned to cultural preferences [29,30]. This stresses the point, already made, that smart sustainable city planners and strategists should prioritise the selection and adaption of culturally aligned technologies, if they are to maximise public acceptance and societal benefit [51,52]. It therefore follows that the potential of smart sustainable city implementation is best understood through the study and analysis of specific urban regions [49,53].

Public acceptance of new technologies is also a function of societal 'maturity', in terms of technological infrastructure and use within the urban environment. This 'maturity' varies widely across the world [54–56], and is sometimes referred to as the 'digital divide' [54,56]. This means that governing authorities within emerging or developing economies face a particular challenge in encouraging the public acceptance of advanced technologies, and particularly those which are smart and sustainable. This is a challenge that can only be met through a more complete understanding of regional culture(s) [45,46,57].

However, the challenge of ensuring that smart sustainable technologies are chosen and adapted to meet local cultural needs is threatened by the prevailing narrative in smart city development. This has evolved over the past two decades, and has resulted in a focus on meta-level issues with smart cities, such as security, ethics and platform interoperability, which have global relevance [12,15,16]. While such a focus is understandable from a historical perspective, recent research suggests that, if maintained, it could result in a standardising effect that represses the necessary understanding of cultural differences that lie behind smart sustainable city success [47,58]. Furthermore, this 'standardising effect' does not seem to be ameliorated by the fact that the concept of the smart city is rapidly being embraced by emerging economies such as India [59]. This is evidenced by the observation that smart developments in these emerging economies seem to align

with the generic strategies of early adopters of smart technology, such as Amsterdam and Barcelona, which did not initially prioritise the sustainability of the planning process, and do not reflect the nuances of local culture [59,60]. This failure to allow local culture to shape smart city development is not a deliberate strategy by the authorities concerned, but a result of a lack of research that shows how culture can impact public acceptance and service effectiveness.

The notion of the citizen-centric development strategy has therefore gained traction in the smart sustainable city research environment [61–63]. However, while there exists considerable research across a number of fields on the subject of technology acceptance, this research requires adaption to meet the specific context of the citizen-centric smart sustainable city [64,65]. We will therefore now examine this study in relation to the extant TAM literature, with a view to the development of the research model of this paper (CITAM).

### 2.1. Citizen-Centric Smart Cities and the TAM

First introduced in 1985 [34], the TAM was originally developed to explain and predict user acceptance of computer technology, specifically focusing on the perceived ease of use and perceived usefulness as key determinants. In 1989, however, the model was revised to account for other factors, such as external variables and social influences, to provide a more comprehensive understanding of technology adoption. Further to this, the Unified Theory of Acceptance and Use of Technology (UTAUT) was introduced in 2003 [66], which integrated elements from several existing models, including the TAM and the Theory of Reasoned Action (TRA). These adaptions include a wide variety of additional factors, such as perceived enjoyment and subjective norms, to enhance the TAM's explanatory power.

In the last few years, however, recent and rapid advances in areas such as the IoT, machine learning and cloud computing have driven further evolution of the TAM to enable a meaningful prediction of how these technologies will be accepted into new markets [67–69]. Despite the sophistication of these models, however, they tend to depend on knowledge generated from the smart city environment, which is mono-dimensional (i.e., technological) in nature, and lacks human, cultural and social attributes [70,71]. It is therefore important to include cultural and social drivers of technology acceptance, to better understand how smart technology can be effectively implemented in different socio-cultural populations [72]. The addition of these factors could have far-reaching effects. Indeed, even though TAMs will remain both relevant and important to smart sustainable city strategies in the context of developed economies, such an advancement to the TAM could have even greater implications for the development of smart sustainable city strategies in emerging economies [72,73].

In order to develop a TAM variation which provides high levels of predictability in the (complex) context of the smart sustainable city, the current study taps into, and builds upon, the history of the TAM [70,71]. We achieve this by combining and refining the factors most relevant to the acceptance of smart technology. These factors were identified by classifying TAMs into categories relevant to citizen-centric smart sustainable city design, as discussed by Attié and Meyer-Waarden [73]. This classification process resulted in the identification of the three principal categories of TAM.

### 2.2. Socially and Commercially Focused Models

This group of models examines technology acceptance through the lens of social, financial and economic influencers. Within this context, social cognitive theory (SCT), introduced by Bandura [74], is particularly relevant, as it can be used to explain how individuals acquire new behaviours and skills through their interactions with the social environment. Widely applied in various fields, SCT provides insights into how self-regulation and self-efficacy play a key role in the development and modification of behaviours such as technology acceptance.

Another important theoretical framework in this category is Diffusion of Innovation theory. Initially developed in 1962, DoIT explains how new ideas, technologies or products spread within a social system over time, by identifying the stages through which an innovation is adopted by members of a community or social group. Updated in 2010, the DoIT helps researchers understand the factors influencing the acceptance and spread of innovations within a given context [75–77]. A further factor that can influence technology acceptance is economic cost. This means that the theory of transaction cost analysis [78] is also relevant to this group of models.

### 2.3. Hedonic Experiential Models

Proposed in 2004, the original hedonic experiential model (HEM) suggests that users' decisions to accept and adopt technology are influenced not only by utilitarian factors, but also by hedonic factors related to the user experience, such as perceived pleasure, aesthetics and novelty [79]. This line of thought was further developed in 2012, when specific hedonic constructs (such as motivation, perceived value, habit and playfulness) were added to the existing Unified Theory of Acceptance and Use of Technology (UTAUT) [80] to produce UTAUT2 [81]. This revised model showed that integrating hedonic elements into TAMs provides a deeper understanding of user behaviour in specific contexts. Other variants, using different psychological constructs, have also been suggested [82], based on the theory of planned behaviour [83].

## 3. Research Model Development and Hypotheses

The concepts of the TAM and SCT have both made important contributions to the field of technology acceptance [84], and both play a key role in the development of the model proposed in this paper.

TAMs have been effectively used in various domains, ranging from ICT and e-commerce to healthcare and education, to understand and predict users' behaviour toward adopting and using new technologies [57,85–87]. The effective application of TAMs across such diverse areas demonstrates their relevance, particularly in domains where understanding user acceptance and adoption is essential for successful technology implementation.

SCT has also been successfully applied in various fields to explain and influence human behaviour. With an emphasis on the interplay between cognitive, behavioural and environmental factors, SCT has proved a valuable framework for understanding complex human behaviours, including technology acceptance, and helping to facilitate positive changes in diverse contexts (e.g., [88–90]).

The proposed model in this study (CITAM) integrates both TAM and SCT to examine users' perceptions and behavioural intention regarding Culturally Informed Technologies (CIT). However, as behaviours towards smart sustainable technologies in a specific context are influenced by many factors, the proposed CITAM framework builds significantly on the two basic constructs of TAM. These constructs, which are "perceived usefulness" (an individual's belief that a technology will be useful for a range of tasks), and "perceived ease of use" (an individual's belief that the technology will be intuitive and simple to apply), are at the core of the established TAM concept [82,85,86]. The model proposed in this study (CITAM) includes these constructs, plus a range of others, identified from the current literature [91–93]. This step was necessary because, although the theories mentioned above (TAM; SCT) have been widely and effectively deployed [85], the literature on either smart cities or smart sustainable cities does not include model adjustments that provide significant insights into user experience and behaviours in the context of Culturally Informed smart technologies [92,94].

In order to identify and integrate relevant constructs into the CITAM framework, a wide range of possibilities were reviewed from the literature. Based on the strength of evidence available, a number (12) of constructs were selected for integration into the CITAM, and associated hypotheses were developed. The added constructs included several which were selected for their influence on behaviour (i.e., Intention to Use) as applied in the Theory of Reason Action [95,96], and were perceived privacy, perceived gains, perceived ease of use, perceived usefulness, replaceability and perceived trustworthiness [97–99].

The current literature suggests that, while perceived privacy is important in all regional and cultural contexts, it is especially important in a smart sustainable city context, where users tend to have lower levels of experience with emerging (smart sustainable) technologies [100], and in cultures where socio-religious values have a strong impact on individual attitudes and behaviours [101]. Perceived privacy is therefore critical to acceptance and adoption [102–104]. It is thus intuitively reasonable to suppose that higher levels of perceived privacy lead to higher intention to use CIT. It is therefore hypothesised that:

**H1:** *Perceived privacy has a positive association with intention to use CIT.*

A second added construct is the metric of perceived gains, which acts as a measure for relating the attractiveness to users of new technologies compared with extant systems [57], and is a strong predictor of technology acceptance. It is therefore hypothesised that:

**H2:** *Perceived gains has a positive association with intention to use CIT.*

Citizens often feel that using new technology requires learning and effort, and this is a particularly strong belief when the technology concerned is related to public services—e.g., government portals [82,105,106]. As there is no evidence that this belief will change for CIT. It is therefore hypothesised that:

**H3:** *Perceived ease of use has a positive association with intention to use CIT.*

Traditional (i.e., non-digital) methods of engaging with public services or dealing with organisations can be difficult and time-consuming in any country, as it can involve time and travel. This tends to be even more true in many emerging economies [100,107]. It is therefore likely that users will be more willing to accept and use a technology if they feel that it offers convenience. It is therefore hypothesised that:

**H4:** *Perceived usefulness has a positive association with intention to use CIT.*

Connectivity is a construct which measures the extent to which the new CIT operates with the user's existing digital ecosystem. This has been shown to be a significant predictor of technology acceptance in all contexts [86]. It is therefore hypothesised that:

**H5:** *Connectivity has a positive association with intention to use CIT.*

Techno-solutions are often thought of as generic concepts that transcend geographical and cultural boundaries. Indeed, some argue that genericity is one of the fundamental characteristics of new technology [108]. An example of this is GPS, which provides location and time information anywhere on Earth, under any weather conditions, at all times. However, there are also many examples of where a technological solution has been adapted to meet the needs of local cultural context. One notable example of this is the use of indigenous knowledge coupled with modern technology to develop sustainable agriculture practices [109–111]. In this context, modern technology can be integrated with local practices to enhance agricultural sustainability while respecting cultural traditions. Mobile apps, for example, can be used for crop management to provide indigenous farmers with information on crop varieties, pest management strategies and weather forecasts

while remote sensing technologies, such as satellite imagery and drones, can be used to monitor soil health and vegetation dynamics in local areas. In this study, perceived cultural adaption refers to the user's perception that a new technology has been sufficiently adapted to their cultural context, and therefore meets their needs precisely, in terms of functionality, sustainability and accuracy in delivery of the required service [97,112,113]. It is therefore hypothesised that:

**H6:** *Perceived cultural adaption has a positive association with intention to use CIT.*

It is known that users tend not to engage with services or systems that they perceive as being inconsistent or unreliable in terms of service quality [74]. Services or systems, on the other hand, which are believed to provide high levels of service quality are significantly more likely to be adopted [114–116]. It is therefore hypothesised that:

**H7:** *Service quality has a positive association with intention to use CIT.*

Digital literacy is a measure of a user's comfort level in using a technology, and their confidence that they can engage with it (the technology) effectively [117]. The greater a user's digital literacy, the more likely they are to accept and adopt a smart sustainable (or any) technology. It is therefore hypothesised that:

**H8:** *Digital literacy has a positive association with intention to use CIT.*

Users often adopt new technologies in order to achieve improvements in working processes on a day-to-day basis [98,118]. If the potential user believes that the technology can deliver such improvements they are significantly more likely to adopt the technology [98,99,118]. It is therefore hypothesised that:

**H9:** *Process improvement has a positive association with intention to use CIT.*

Economic advantage and convenience can also play a key role in influencing technology adoption. These can be achieved in a variety of ways, and particularly through savings on cost [85,86], energy and time [85,86,119–121]. Thus, the constructs of cost saving, energy efficiency and time efficiency all have a positive association with intention to use CIT. It is therefore hypothesised that:

**H10:** *Cost saving has a positive association with intention to use CIT.*

**H11:** *Energy efficiency has a positive association with intention to use CIT.*

**H12:** *Time efficiency has a positive association with intention to use CIT.*

Figure 1 shows the structure of the proposed CITAM, covering TAM and SCT concepts, as well as the relationship between constructs.

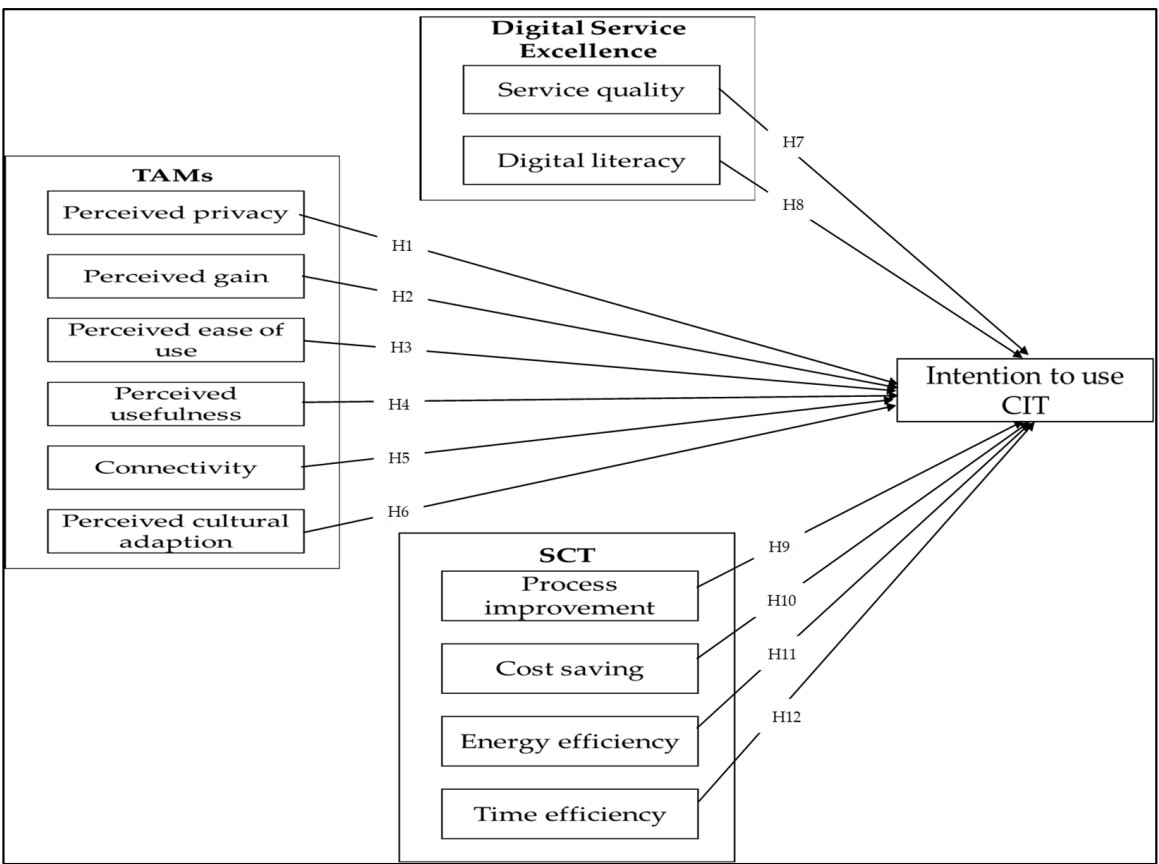

**Figure 1.** The proposed research model.

## 4. Methodology

### 4.1. Survey Development

This study used a survey approach to examine the hypotheses in the research model. The survey used a standard 5-point Likert scale to measure responses, and included 43 items to evaluate the 12 constructs (Table 1). While the survey included some items that were adapted from other studies, most were specifically designed, following standard guidelines [122,123], to meet the needs of this research.

As the survey included adapted and new items, its content validity was assessed before the data collection phase [124–126]. To achieve this, the researchers consulted with a number of experts in the field, specifically chosen for their experience and research credentials [127]. While there is no universally accepted 'rule' for how many experts this process requires, a minimum of three is generally used [128]. We therefore reached out to 8 experts, and received feedback from 5. Due to this feedback, the original set of 50 items were reduced to 43.

These items were then reviewed again for validity and accuracy, and a pilot study was carried out to evaluate the clarity and effectiveness of the questionnaire. This pilot involved a group of 30 respondents, with a representative demographic cross-section, and resulted in various adjustments to the items to eliminate some identified ambiguities and enhance response accuracy. This helped to ensure the quality of the data collected for analysis.

**Table 1.** Constructs and their definitions and items factor loadings.

| Construct | Items | Loading |
|---|---|---|
| **Perceived privacy (The perception that CIT will protect personal data)** | | |
| PP1 | CIT meets my specific security needs | 0.876 |
| PP2 | I feel confident that CIT will adequately protect me from security threats on the Internet. | 0.815 |
| PP3 | I feel confident using CIT for financial transactions. | 0.848 |
| PP4 | The CIT platform owners will compensate me for any problems I have using their system. | 0.849 |
| **Perceived gains (The perception that CIT will offer advantages over its predecessors)** | | |
| PG1 | I feel that using CIT will be an improvement on my current systems. | 0.851 |
| PG2 | CIT perfectly fits the way I like to work. | 0.921 |
| PG3 | Working with CIT is generally considered to be a good idea. | 0.830 |
| PG4 | I believe that CIT is a good solution for most citizens. | 0.756 |
| **Perceived ease of use (The perception that CIT will be intuitive and easy to use)** | | |
| PEOU1 | I believe that using CIT will be straightforward. | 0.931 |
| PEOU2 | I have no concerns about using CIT. | 0.903 |
| PEOU3 | Most people would be comfortable using CIT. | 0.732 |
| PEOU4 | I feel that CIT is designed to be simple and intuitive. | 0.951 |
| **Perceived usefulness (The perception that CIT will improve performance in given tasks)** | | |
| PU1 | CIT would make daily life easier for me. | 0.868 |
| PU2 | CIT offers many advantages for everyday tasks. | 0.839 |
| PU3 | I would find CIT useful. | 0.896 |
| PU4 | I feel that using CIT puts me in control of my affairs. | 0.878 |
| **Connectivity (The perception that CIT easily connects to other systems)** | | |
| CO1 | CIT lets me work from any city. | 0.728 |
| CO2 | CIT is highly compatible with other systems I use. | 0.742 |
| CO3 | When using CIT, interacting remotely with other systems and people is easy and reliable. | 0.867 |
| CO4 | CIT gives me the freedom to work when and where I want. | 0.836 |
| **Perceived cultural adaption (The perception that CIT reflects the nuances of cultural needs)** | | |
| PCA1 | Using CIT means that I can work in precisely the way I like to work. | 0.780 |
| PCA2 | I believe that CIT provides functionality which meets my needs exactly. | 0.732 |
| PCA3 | I feel that CIT guides and manuals are very easy to understand and follow. | 0.795 |
| PCA4 | I feel that the developers of CIT fully understand what I need to work effectively. | 0.801 |
| PCA5 | I believe that working with CIT is just as effective as being in the physical workplace. | 0.734 |
| **Quality of service (The perception that CIT will provide the standards of service required)** | | |
| QS1 | Using CIT gives me confidence that I can perform my work efficiently. | 0.862 |
| QS2 | I feel that CIT has powerful data protection and backup features. | 0.858 |
| QS3 | CIT helps to eliminate human error. | 0.878 |
| **Digital literacy (The user's confidence in their ability to understand and work with IT)** | | |
| DL1 | I am confident that I can use the CIT features effectively. | 0.921 |
| DL2 | I have no concerns about learning to use all of the CIT features. | 0.923 |
| DL3 | I feel that I have no requirement for support in using CIT. | 0.869 |
| **Process improvement (The perception that CIT will make improvements to daily processes)** | | |
| PI1 | CIT will improve the way I work and the results I produce. | 0.921 |
| PI2 | I believe my employer will benefit from my greater efficiency, due to CIT. | 0.881 |
| PI3 | Tasks are easier and faster when using CIT. | 0.912 |
| **Cost savings (The perception that CIT will deliver financial efficiencies)** | | |
| CS1 | CIT reduces the cost of administrative processes. | 0.754 |
| CS2 | Using CIT results in lower travel costs. | 0.887 |
| CS3 | CIT helps people work more efficiently and cost-effectively. | 0.843 |
| **Energy efficiency (The perception that CIT will deliver energy savings)** | | |
| EE1 | Using CIT reduces travel, and therefore fuel consumption. | 0.901 |
| EE2 | CIT allows remote working, which reduces the energy costs associated with running offices. | 0.779 |
| **Time efficiency (The perception that CIT will deliver time savings)** | | |
| TE1 | CIT reduces the time I need to complete most tasks. | 0.806 |
| TE2 | I believe that using CIT saves me time by reducing my need to travel. | 0.921 |
| TE3 | CIT helps people match their working hours to their specific needs. | 0.856 |
| **Behavioural intention** | | |
| BI1 | I will continue to use CIT. | 0.893 |
| BI2 | I intend to increase my use of CIT when I take on new tasks. | 0.865 |
| BI3 | I speak highly of CIT and advise others to engage with it. | 0.931 |

### 4.2. Sample and Data Collection

All participants in this study (N = 661) were residents of Saudi Arabia, which included participants with a range of nationalities, ages, private/public sector occupations, and socioeconomic statuses. The data was collected using Google Forms, and recruitment strategies included social media outreach, community and association centres, and email invitations sent via collaborating public bodies (i.e., organisations with an interest in the results of the research).

Over a four-month period, a total of 744 responses were received, of which 121 were considered invalid due to incomplete, ambiguous or inconsistent survey completion, resulting in 661 valid responses for analysis. Table 2 shows a summary of the respondents' demographic profiles.

**Table 2.** Demographic Characteristics of Survey Participants.

| Demographic Profile | | Participants % |
|---|---|---|
| Gender | Male | 58 |
| | Female | 42 |
| Education | Bachelor | 43 |
| | Master | 39 |
| | PhD | 18 |
| Age | 18–30 | 36 |
| | 31–40 | 32 |
| | 41–50 | 23 |
| | 51+ | 9 |
| Nationalities | Saudi | 58 |
| | India | 16 |
| | Egypt | 7 |
| | US | 9 |
| | UK | 5 |
| | France | 3 |
| | China | 2 |

### 4.3. Ethical Considerations

All relevant ethical guidelines were observed during the data-gathering process. Prior to participation, all respondents were fully informed about the study's purpose and procedures, and were asked to provide informed consent, which could be withdrawn at any time, for any reason. All respondents were also assured of total anonymity, and all participation was fully voluntary, with no incentive (financial or otherwise), offered to any participant. The research protocol, including the participant recruitment strategy, received full approval from the University's Research Ethics Board (REB).

## 5. Results and Analysis

The model proposed in this paper was assessed by testing both validity and reliability. This was achieved by using SmartPLS 4.0 software [129,130] to apply SEM (structural equation modelling). The results of this analysis will indicate the accuracy of the suggested relationship between the constructs and their dependent terms. Below, we describe the results of this process.

### 5.1. Convergent Validity

Convergent validity is a critical component in establishing overall construct validity, which evaluates the extent to which a measurement instrument accurately assesses the underlying theoretical construct it is intended to measure [131]. In order to determine convergent validity, various measures (factor loading, Cronbach's alpha (CA), Average Variance Extracted (AVE), and Composite Reliability (CR)) are used [130,132].

For a construct to be strongly associated with an associated item or variable, the factor loading for that association should be above 0.7 [133], and it can be seen from Table 1 that this was the case for all items and their related construct in the model proposed by this study.

The values of the other three indicators of convergent validity, calculated from the data, are shown in Table 3. CA, which is an important measure of internal consistency [131], was found to be well above the recommended value of 0.7 [134], and this was supported by the values of the CR coefficient, which were also greater than the recommended value of 0.7 [131]. Given that the AVE values for all constructs were greater than 0.7, which is above the commonly accepted threshold (0.5) for good convergent validity [133], the results for all criteria were considered to be satisfactory.

**Table 3.** Summary of Construct Reliability and Validity Measures.

| Construct/Factor | CA | CR | AVE | Correlations | | | | | | | | | | | | |
| | | | | 1 | 2 | 3 | 4 | 5 | 6 | 7 | 8 | 9 | 10 | 11 | 12 | 13 |
|---|---|---|---|---|---|---|---|---|---|---|---|---|---|---|---|---|
| Perceived privacy | 0.83 | 0.84 | 0.85 | **0.93** | | | | | | | | | | | | |
| Perceived gains | 0.81 | 0.83 | 0.82 | 0.62 | **0.91** | | | | | | | | | | | |
| Perceived ease of use | 0.83 | 0.84 | 0.74 | 0.68 | 0.69 | **0.86** | | | | | | | | | | |
| Perceived usefulness | 0.85 | 0.79 | 0.72 | 0.55 | 0.63 | 0.68 | **0.85** | | | | | | | | | |
| Connectivity | 0.81 | 0.77 | 0.74 | 0.57 | 0.69 | 0.6 | 0.56 | **0.86** | | | | | | | | |
| Perceived cultural adaption | 0.80 | 0.82 | 0.80 | 0.55 | 0.66 | 0.61 | 0.6 | 0.53 | **0.89** | | | | | | | |
| Quality of service | 0.83 | 0.74 | 0.84 | 0.62 | 0.69 | 0.45 | 0.66 | 0.58 | 0.55 | **0.92** | | | | | | |
| Digital literacy | 0.84 | 0.72 | 0.79 | 0.48 | 0.5 | 0.37 | 0.56 | 0.47 | 0.45 | 0.82 | **0.89** | | | | | |
| Process improvement | 0.72 | 0.83 | 0.77 | 0.72 | 0.56 | 0.74 | 0.45 | 0.67 | 0.49 | 0.75 | 0.62 | **0.88** | | | | |
| Cost savings | 0.74 | 0.85 | 0.83 | 0.73 | 0.6 | 0.79 | 0.44 | 0.73 | 0.54 | 0.78 | 0.76 | 0.6 | **0.91** | | | |
| Energy efficiency | 0.85 | 0.83 | 0.81 | 0.67 | 0.77 | 0.67 | 0.61 | 0.48 | 0.5 | 0.37 | 0.56 | 0.47 | 0.45 | **0.90** | | |
| Time efficiency | 0.82 | 0.85 | 0.83 | 0.60 | 0.86 | 0.67 | 0.76 | 0.51 | 0.89 | 0.63 | 0.71 | 0.68 | 0.57 | 0.62 | **0.91** | |
| Behavioural intention | 0.74 | 0.82 | 0.82 | 0.64 | 0.66 | 0.6 | 0.58 | 0.73 | 0.54 | 0.78 | 0.76 | 0.66 | 0.6 | 0.58 | 0.66 | **0.91** |

Note: the square root of AVE is shown in bold.

### 5.2. Discriminant Validity

Discriminant validity ensures the distinctiveness of each construct, affirming its individual identity [135,136]. One of the most common techniques used to check discriminant validity is the Fronell–Larcker criterion [133]. According to this criterion, the square root of the average variance extracted by a construct must be greater than the correlation between the construct and any other construct. The Fronell–Larcker criterion was therefore evaluated by comparing the correlation coefficients of constructs. As can be seen in Table 3, the construct values are more than the correlation between them, supporting the conclusion that all latent variables differ sufficiently from each other [135,136].

### 5.3. Structural Model

The construct validity of the model's factors was tested through confirmatory factor analysis (CFA) using SEM. The R-Squared values for the constructs were examined to determine the effect sizes, as recommended by Hair et al. [132], and these can be seen (Table 4) to be of sufficient size to support the hypotheses [137], while the R-Squared values suggest that the model has an acceptable fit. The R-Squared value for behavioural intention (0.879) and the standardised path coefficients also suggest a strong relationship between the dependent constructs and BI.

**Table 4.** Analysis of hypotheses testing results: Standardized Coefficients, Significance, and Model Diagnostics.

| Hypothesis | β (Beta) | Std. Coefficient | *t*-Value | Decision | f² | R² | VIF | Q² |
|---|---|---|---|---|---|---|---|---|
| H1 | 0.101 | 0.042 | 4.176 | Supported *** | 0.154 | 4.125 | 0.654 | 0.879 |
| H2 | 0.108 | 0.051 | 1.849 | Supported * | 0.051 | 4.354 | | |
| H3 | 0.113 | 0.059 | 1.864 | Supported * | 0.022 | 4.436 | | |
| H4 | 0.128 | 0.035 | 2.426 | Supported * | 0.089 | 4.128 | | |
| H5 | 0.106 | 0.046 | 1.614 | Supported * | 0.039 | 3.711 | | |
| H6 | 0.168 | 0.089 | 1.811 | Supported *** | 0.014 | 4.877 | | |
| H7 | 0.065 | 0.032 | 2.870 | Supported ** | 0.110 | 3.620 | | |
| H8 | 0.098 | 0.033 | 4.414 | Supported ** | 0.158 | 3.964 | | |
| H9 | 0.098 | 0.061 | 6.069 | Supported ** | 0.385 | 4.079 | | |
| H10 | 0.088 | 0.032 | 4.245 | Supported ** | 0.115 | 3.677 | | |
| H11 | 0.035 | 0.058 | 0.415 | Not supported | 0.001 | 2.889 | | |
| H12 | 0.098 | 0.057 | 1.913 | Supported * | 0.080 | 4.878 | | |

Note: * for $p < 0.05$, ** for $p < 0.01$, *** for $p < 0.001$.

To test for the presence of multicollinearity, variance inflation factor (VIF) values were examined. As the VIF analysis showed all values below three, which is generally considered to suggest an acceptably low correlation between variables [130,132], multicollinearity was not considered to be a significant issue. The model's predictive relevance was further evaluated using blindfolding [130,132]. The values of Q-Squared for BI (0.641) are significantly greater than zero, suggesting that the model shows moderately high predictive performance. Finally, *t*-values were used to assess the relationship between constructs and BI [130,132]. As is summarised in Table 4, all hypotheses, with the exception of H11, were supported.

*5.4. Model Fit*

The final step in assessing the CITAM proposed in this study was to calculate its Goodness of Fit index, a process which is frequently used to evaluate how well a statistical model, particularly an SEM model, fits the observed data [138]. The Goodness of Fit index for CITAM was obtained by following the method described by Sohaib [139]. The Goodness of Fit of the CITAM was found to be 0.77, which suggests a good fit—i.e., that the model explains a large proportion of the variance in the observed data. Table 5 shows all indices for model fit.

**Table 5.** The model fit indices.

| Fit Measure Category | Fit Measure | Result | Meets Recommended Criteria? |
|---|---|---|---|
| Absolute fit measures | Chi-Square ($\chi^2$/DF) | 2.320 | Yes (<3.0) |
| | SRMR | 0.962 | Yes (>0.80) |
| | GFI | 0.978 | Yes (>0.90) |
| | RMSEA | 0.043 | Yes (<0.05) |
| Parsimonious fit measures | PGFI | 0.652 | Yes (<0.05) |
| | PNFI | 0.671 | Yes (<0.05) |
| Incremental fit measures | AGFI | 0.938 | Yes (>0.90) |
| | IFI | 0.951 | Yes (>0.90) |
| | NFI | 0.951 | Yes (>0.90) |
| | CFI | 0.965 | Yes (>0.90) |

## 6. Discussion

The aim of this study was to develop a predictive model for smart sustainable technology acceptance that takes account of regional cultural context. This moves the current literature forward in significant ways. Firstly, most existing studies implicitly treat the smart sustainable city as a generic technological model, conforming to a concept derived from the developed world. However, many smart cities are now being planned in developing economies, which require the concept of the smart sustainable city to be defined and shaped according to local and regional culture [91–93,140]. Secondly, previous studies have built their models around TAMs which are limited in their range of constructs, and which focus on the fundamental, but generic, constructs of usefulness and ease of use [117,141]. This study, however, integrates a wider range of constructs into its model. Another important difference between this study and previous research is that the current literature has tended to focus on the technological infrastructures of e-government [64,65], and have not fully considered the wide range of culturally nuanced services associated with smart sustainable cities.

In order to develop a model which meets the needs of this study, constructs from the original TAM model, proposed by Davis et al. [34], were integrated with constructs from SCT [74]. The resulting structural framework allowed for the examination of how a citizen-centric, culturally informed approach to the development of smart sustainable technology affects acceptance behaviours within a specific cultural context (in this case, Saudi Arabia). Such a citizen-centric approach has largely been overlooked in the recent literature related to smart cities [117,141].

The current study proposed twelve hypotheses relating various constructs to behavioural intention. An analysis of the data collected by the study's survey showed that only one of these hypotheses was not supported. This was H11, which proposed that cost savings would be a positive driver of intention to use CIT. All other factors, such as digital literacy, process improvement, and—importantly—perceived cultural adaption (which includes sustainability expectations)—had a clear and significant positive effect on CIT. These results are interesting, as it is intuitively tempting to assume that cost saving would be a notable incentive to adopt smart technology [91,92], and the results of some studies, in other technological contexts, suggest that the financial savings from replacing legacy systems with new technology can affect adoption and acceptance [142]. The results of this study, however, suggest that, in the context of the smart sustainable city, the perceived benefit of cost savings is eroded, and perhaps erased, by other factors, such as the cultural adaption and alignment of the new technology, and improved personal efficiency. This finding accords with other studies that have used SCT to examine user behaviour in other contexts [117,141,143], and found that an individual's sense of familiarity and self-confidence in using a technology can override other factors in terms of acceptance.

Although the studies just mentioned were not conducted in the specific context of the smart, or smart sustainable, city, it is worth noting that there exist some studies (e.g., [144,145]) that do focus on smart cities, and which have reported similar findings—i.e., that technology acceptance tends to be more influenced by individual 'comfort' factors, such as security and ease of use, than utilitarian factors such as cost savings. This suggests that there is potential for cultural adaption of technology to play a significant role in the acceptance and adoption of smart sustainable technology.

This suggestion is supported by the findings of this paper, which are based on data collected from Saudi Arabia. These findings are useful from both a theoretical and a practical perspective. From the theoretical angle, the paper's validated model could be used by city planners and strategists, as well as other stakeholders, to assess the potential for acceptance of a particular smart service or technology, before investing in its development and subsequent implementation. From a practical perspective, carrying out such a pre-development assessment is entirely feasible and relatively straightforward, as the public perceptions of the various constructs can be measured through the use of surveys.

As has been noted above, the aim of the model proposed in this study (the CITAM) was to provide higher predictive power, in terms of the acceptance and adoption of smart sustainable city technologies, than previous models, which include the general TAM [34,146,147] and the Unified Theory of Acceptance and Use of Technology (UTAUT) [82]. The various tests on the validity of this study's model (described in Section 5) have confirmed that it succeeds in this aim. It does so by integrating a wider range of factors, such as digital literacy, cultural adaption and connectivity, and demonstrating that these factors are (either consciously or subconsciously) prioritised by users during the decision-making process for acceptance. This is consistent with the findings of other research [92,93] which found that a variety of factors are influential in the general issue of technology adoption.

This paper makes a useful contribution to both research and practice in the field of technology acceptance. One of these contributions is that the study adds to the current literature through the introduction of the CITAM. Whereas previous research, in the context of smart cities, has used acceptance models which tend to have a relatively narrow focus on utilitarian factors [91–93], CITAM uses a citizen-centric model which integrates a wider range of factors, including the perception of (perceived) cultural adaption. The centrality of a citizen-centric approach in the success of an acceptance model has been emphasised by Wu et al. [92]. While cultural adaption was shown by this study to be important in influencing smart sustainable technology acceptance, other factors were also important, such as perceived security, perceived gains, connectivity and digital literacy.

The paper's focus on the context of Saudi Arabia is another enhancement of the current literature, as Saudi has two important characteristics which make its national context unique and important. The first is the Saudi Vision 2030 [35,36], in which smart cities and sustainability play a key role [36]. Secondly, the Saudi population is predominantly (90%) Arab, and most of the remainder is Afro-Asian [35–37], which means that it is culturally significantly different from the Western developed populations from which the generic model of the smart city and the smart sustainable city has evolved. Taken together, these two factors give Saudi Arabia rich potential for examining the impact of CIT on acceptance and adoption behaviour.

## 7. Conclusions, Limitations and Future Research

The objective of this study was to develop a more citizen-centric model for smart sustainable technology acceptance, by understanding the impact of cultural factors, and particularly the extent to which the technology is adapted to cultural requirements. The results of the study showed that the use of CIT has significant potential for increasing sustainable technology acceptance in a smart city context. By using the model, city planners and strategists can meaningfully assess, using survey instruments, the importance of culturally adapted smart sustainable technologies intended for public services before investing in development and implementation. Until now, no other model has emerged which allows such assessment with the accuracy and sensitivity offered by the model proposed in this study. The findings will therefore help to improve the process of transitioning to a smart sustainable city, by allowing authorities to understand the significance of cultural adaption as well as the relative importance of a range of factors which are a function of cultural context.

It should be noted, however, that the research has some limitations. One of these limitations is that, although the CITAM framework is based upon proven theoretical constructs, such as the TAM and SCT, it is nonetheless a new and untested concept. The alignment and integration of the culturally informed dimension may not lead to general applicability of the model. Future research could usefully test this. Secondly, this research, as a citizen-centric study, focuses (by definition) on the perceptions, attitudes and behaviours of individuals. However, given that the success of smart cities also depends on acceptance at the organisational level, future research could usefully examine how CIT impacts the behaviour of organisations and enterprises. Thirdly, the validity of the research model is based on a sample that comprised individuals within a mainly Arab culture. Testing

the model's validity in the context of other cultures would further enhance the literature. It would be particularly interesting to test the model's validity in the context of Western culture, which has its own internal subcultures and variants, and is significantly impacted by other global cultures.

Additionally, the research model of this study relies on cross-sectional data, indicating that the findings reflect correlations rather than causality. Future research could use longitudinal data to expand the current model, offering a more precise perspective on the dynamics of smart city adoption over time. Lastly, this study mainly relies on quantitative research, which may not fully capture the nuanced and detailed insights needed to understand the complex dynamics of smart city adoption. Therefore, future research in the field of smart cities and digital transformation should consider incorporating qualitative approaches for a deeper and more comprehensive understanding of these multifaceted phenomena.

In summary, the study provides a practical and powerful tool (the CITAM) that helps smart city planners understand the importance of the perception, by citizens, of the cultural alignment of proposed smart sustainable technologies. This will give planners and decision makers confidence that they have taken appropriate steps, in terms of the cultural adaption of proposed technologies, to maximise acceptance. The practical benefits of this, such as cost-efficiency, is high. An even greater motivation for deploying CITAM is the many benefits that accrue, at a societal level, from the enthusiastic engagement of citizens with efficient public services.

**Funding:** This research was funded by the Researchers Supporting Project number (RSP2024R233), King Saud University, Riyadh, Saudi Arabia.

**Institutional Review Board Statement:** The study was carried out in accordance with the principles outlined in the Declaration of Helsinki and received approval from the Institutional Review Board (Human and Social Research) at King Saud University.

**Informed Consent Statement:** All participants involved in the study provided informed consent.

**Data Availability Statement:** Data can be made available upon request to ensure privacy re-strictions are upheld.

**Acknowledgments:** The author would like to extend his sincere appreciation to the Researchers Supporting Project (RSP2024R233), King Saud University, Riyadh, Saudi Arabia.

**Conflicts of Interest:** The author declares no conflicts of interest.

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
