# Peer review of "Culturally Informed Technology: Assessing Its Importance in the Transition to Smart Sustainable Cities"

_sustainability, doi:10.3390/su16104075_

Round 1
Reviewer 1 Report
Comments and Suggestions for Authors
Author has prepared article of high quality. The principal aim of author study is to help smart sustainable city planners and strategists, as well as industries and businesses, understand the importance of ‘Culturally Informed’ technologies which are appropriate to the local context. Author in a good quality elaborated in section 3 his Research model development and hypotheses. A solid results summary and overall analysis has been performed in section 5. A question for consideration would be if the section 4 methodology should follow section 3. Research model development and hypotheses or if it should not preceed the section 3. Overall it can be stated that author paper makes a useful contribution to both research and practice in the field of technology acceptance.
Author Response
Dear Reviewer,
Thank you again for your time and effort in reviewing my work. I have incorporated your feedback and believe that our revised version effectively addresses the concerns raised (please see the attached revised Manuscript and respond letter).
I look forward to any additional guidance you may provide during the final stages.
Sincerely,
Authors

Reviewer 2 Report
Comments and Suggestions for Authors
Dear Authors,
the manuscript is up-to-date and scientifically treats the topic of smart cities from several points of view in an appropriate way. However, after careful inspection, I have a few reservations:
- do not use abbreviations in the abstract (line 15 and 16),
- sort keywords alphabetically, also keywords do not have to start with a capital letter,
- the seventh chapter marked as Conclusion is relatively short; in accordance with the instructions for the authors, I recommend expanding it with several paragraphs devoted to the limits of this research and possibly also to the issue of further research into the creation and construction of smart cities.
Considering the number of scientific sources used, it is clear that this scientific work has a very good theoretical basis, but in the list of references on pages 15-20, more attention should be paid to the correct labeling of the individual sources used, for example:
- references 7, 8, 23 without web links,
- references 52 and 131 are duplicates,
- reference 80 - unidentifiable names of authors,
- references 94,95,96 are duplicates,
- reference 105 - there is no reason for it to be written in capital letters,
- reference 140 - other identification data of this work are missing.
These comments or in my opinion, their inclusion is not difficult, but also necessary for the formal and visual aspects of this manuscript.
Good luck Reviewer
Author Response

(The authors gave the same response as above.)

Reviewer 3 Report
Comments and Suggestions for Authors
The article presents the Culturally Informed Technology (CIT) Acceptance Model (CITAM), a novel framework aimed at integrating cultural factors into the evaluation of public acceptance of smart sustainable technologies. The model's validation was conducted through a survey administered in various cities across Saudi Arabia.
The incorporation of cultural considerations into the assessment of technology acceptance is commendable, particularly in the context of smart city initiatives. However, the introduction appears somewhat broad and could benefit from a more focused discussion of the smart city context. The understanding of a smart
For instance, while examples like WeChat and Alipay are mentioned, they may not be the most suitable illustrations for the smart city context, given their applicability to a vastly populous country like China.
A fundamental concern arises regarding the assumption that smart cities are solely reliant on smart and sustainable technologies, thus implying the suitability of the Technology Acceptance Model (TAM). In reality, smart city initiatives prioritize technology-driven solutions that address specific challenges faced by citizens. Moreover, cultural differences significantly influence the acceptance of technology, as evidenced by varying attitudes towards innovations like WeChat and video surveillance with face recognition across different regions. The invisibility of certain technologies, such as blockchain, further complicates research on technology acceptance, as citizens may not be aware of the underlying technology behind a service.
While the TAM serves as a foundational model for the study, its relevance to the research topic should have been thoroughly discussed beforehand.
Despite these reservations, the research demonstrates robust methodology.
The results lack a comprehensive discussion on the general applicability of the CITAM model. As a resident of a smart city in Europe, I remain unconvinced of the model's suitability in this context.
In conclusion, while the research is methodologically sound, further refinement is needed to address concerns regarding the applicability and validity of the CITAM model, particularly in diverse cultural and geographical settings beyond Saudi Arabia.
Comments on the Quality of English Languagen/a
Author Response

(The authors gave the same response as above.)
